# Sentinel Lymph Node Biopsy in Endometrial Cancer: Dual Injection, Dual Tracer—A Multidisciplinary Exhaustive Approach to Nodal Staging

**DOI:** 10.3390/cancers14040929

**Published:** 2022-02-13

**Authors:** Anna Torrent, Joana Amengual, Catalina Maria Sampol, Mario Ruiz, Jorge Rioja, Gabriel Matheu, Pilar Roca, Octavi Cordoba

**Affiliations:** 1Gynecologic Oncology Unit, Obstetrics and Gynecology Department, Hospital Universitari Son Espases, 07120 Palma, Spain; joana.amengual@ssib.es (J.A.); mario.ruiz@ssib.es (M.R.); jorgeo.rioja@ssib.es (J.R.); octavi.cordoba@ssib.es (O.C.); 2School of Medicine, Universitat de les Illes Balears (UIB), 07120 Palma, Spain; 3Department of Nuclear Medicine, Hospital Universitari Son Espases, 07120 Palma, Spain; catalinam.sampol@ssib.es; 4Department of Pathology, Hospital Universitari Son Espases, 07120 Palma, Spain; gmatheu@ssib.es; 5Department of Biology and Molecular Biology, Universitat de les Illes Balears (UIB), 07122 Palma, Spain; pilar.roca@uib.es; 6Institut d’Investigació Sanitària de les Illes Balears, IdISBa, 07120 Palma, Spain

**Keywords:** endometrial cancer, sentinel lymph node, learning curve, indocyanine green, radiotracer, lymphatic mapping

## Abstract

**Simple Summary:**

Since clinical guidelines accepted the utilization of sentinel lymph node (SLN) in apparent uterine-confined endometrial cancer (EC), many centers have already used it as a standard staging surgery. The most widely accepted tracer is ICG alone, but several studies comment on the importance of surgeon experience in order not to lose sensitivity in the first 30-40 cases. This is a prospective, observational single-center trial designed to improve SLN detection rate specially during learning curve. The application of dual tracer (indocyanine green (ICG) and Technetium99 (Tc99) injected separately) helps us to reach a very good overall and bilateral SLN pelvic detection rate in early-stage endometrial cancer patients. Dual injection (cervical and fundal) improves para-aortic SLN detection rate.

**Abstract:**

Introduction: Sentinel lymph node (SLN) has recently been introduced as a standard staging technique in endometrial cancer (EC). There are some issues regarding team experience and para-aortic detection. Objective: to report the accuracy of SLN detection in EC with a dual tracer (ICG and Tc99) and dual injection site (cervix and fundus) during the learning curve. Methods: A prospective, observational single-center trial including 48 patients diagnosed with early-stage EC. Dual intracervical tracer (Tc99 and ICG) was injected at different times. High-risk patients had a second fundus injection with both tracers. Results: the detection rates were as follows: 100% (48/48) overall for SLNs; 98% (47/48) overall for pelvic SLNs; 89.5% (43/48) for bilateral SLNs; and 2% (1/48) for isolated para-aortic SLNs. In high-risk patients, the para-aortic overall DR was 66.7% (22/33); 60.7% (17/28) with ICG and 51.5% (17/33) with Tc99 (*p* = 0.048)). Overall rate of lymph node involvement was 14.6% (7/48). Macroscopic pelvic metastasis was found in four patients (8.3%) and microscopic in one case (2%). No metastasis was found in any para-aortic SLNs. Half of the patients with positive pelvic SLNs had positive para-aortic nodes. In high-risk patients, when para-aortic SLNs mapped failed, 36.4% (4/11) had positive nodes in para-aortic lymphadenectomy. The sensitivity and negative predictive value (NPV) of SLN pelvic detection was 100%. Conclusions: Multidisciplinary exhaustive approach gives a suitable accuracy of SLN during learning curve. Dual injection (cervical and fundal) with dual tracer (ICG and Tc99) offers good overall detection rates and increases para-aortic SLN detection.

## 1. Introduction

Endometrial cancer (EC) is the most common female genital malignancy with rapidly increasing incidence, especially in developed countries [1]. Lymph node status remains one of the most important prognostic factors, and is an essential part of staging surgery in pre-operative early EC, as lymph nodes metastasis will determine adjuvant therapy. The overall rate of nodal involvement in stage I is in the range of 10–17% [2,3].

European and American guidelines recently accepted sentinel lymph node (SLN) biopsy for staging purposes in patients with FIGO stage I–II disease [4,5]. The objective of SLN mapping is to secure information concerning lymph node status, yet minimize collateral damage if complete lymphadenectomy can be avoided. Most importantly, it seems to not compromise the survival outcomes of EC. Many studies have shown good sensitivity and low false positive rates [6].

Superficial and deep cervical injection with indocyanine green (ICG) tracer is the more accepted detection technique. However, cervical injection alone is associated with a significantly lower rate of aortic detection compared with uterine injection [7]. The possibility of missing occult para-aortic lymph nodes metastasis is from 1% to 3% [8,9]. Another concern is if the effort to obtain para-aortic detection, independently of pelvic mode status, gives us enough clinical value.

Most guidelines insist on surgeon experience for ICG technique, and it has been demonstrated that the diagnostic accuracy of SLN increases with surgical team experience [10].

The aim of this study was to evaluate the performance of dual tracer and dual injection technique (ICG and Tc99 into the cervix and uterine fundus) in patients with EC treated in our institution during our learning curve of SLN biopsy. Secondary objectives were to document the feasibility, sensitivity, and NPV of the procedure.

## 2. Materials and Methods

This study was a prospective, observational single-center trial including 48 patients with histologically confirmed EC in apparently pre-operative FIGO 2009 early-stage [11] who underwent minimally invasive surgery between December 2018 and December 2020. The exclusion criteria are shown in Table 1.

All patients signed a specific consent form. The present study was approved by the local Ethics Committee of Balearic Islands, Spain (CEI-IB. Ref. IB 4103/20 PI).

The pre-operative assessment included transvaginal ultrasound and magnetic resonance imaging to assess myometrial invasion. In high-risk patients, a thoracoabdominal CT was performed.

According to pre-operative staging, patients were divided into three groups according to the ESGO–ESTRO-ESP new recommendations of prognostic risk groups [4]. Patients were assessed by a multidisciplinary tumor board and surgery was planned according to her classification as shown in Table 2.

### 2.1. Injection Technique

Radiotracer injection: Injected 24–3 h before surgery. Cervical injection: 2 mL-4 mCi 99mTc-nanocolloid (NanoHSA, ROTOP, Curium Pharma, Berlin, Germany) with four injections through a 22G needle. A total of 0.5 mL at 3 and 9 o’clock superficially (2–3 mm) and 0.5 mL deep (10 to 15 mm cervical depth). For myometrial-fundal injection, we used the TUMIR technique (transvaginal ultrasound-guided myometrial injection of radiotracer), described by Torne et al. [12]. A total of 222 MBq-8 mL 99m Tc-nanocolloid was injected using a caliber 17G biopsy needle of 300 mm length (Vitrolife Sweden, Ref. 17812). During the procedure, 2 mL of endovenous midazolam were administrated for improving patient experience. Pre-operative imaging with single photon emission computed tomography (SPECT/CT) was performed at 1–2 h after the injection of the Tc99 tracer and were discussed before the surgery in a multidisciplinary assessment of the nuclear medicine and the surgery team. (Figure 1).

ICG injection: These injections were performed after anesthetic induction and trocar introduction, allowing direct observation of ICG migration. On high-risk patients we performed a fundal injection (Figure 2), with 4 mL of diluted ICG injected into the myometrial fundus by passing a 17 G needle through the cervical orifice following the technique described by Ruiz et al. [13]. For cervical injection, 4 mL of diluted ICG (25 mg ICG-5 mg/mL in 10 mL sterile water) were injected at the 3 and 9 o’clock positions: 1 mL at 2 to 3 mm cervical depth and 1 mL at 10 to 15 mm depth. Once fundal and cervical injection were performed, surgeon explored the para-aortic area first and then the pelvic area.

Intra-operative SLN detection: In high-intermediate patients, the para-aortic area was explored first once fundal injection had been performed. The peritoneum was opened at the level of the aortic bifurcation. ICG migration was observed, and the gamma probe was toured from aortic bifurcation to the left renal vein. If there was ICG or Tc99 catchment, SLNs were biopsied. Gamma probe verification was performed with the SLN out of the surgical field to check the intensity of Tc99 catchment. Next, systematic para-aortic lymphadenectomy was performed from the aortic bifurcation until reaching the left renal vein as the upper limit of dissection. Then, the pelvic area was explored, with the same exploring protocol as for the para-aortic area. Once SLNs were biopsied, a pelvic lymphadenectomy was performed in intermediate- and high-risk patients.

### 2.2. Histological Analysis

SLNs were sent for intra-operative frozen section. An ultra-staging protocol was performed in most patients (93.7%). Two paraffin-embedded slides were created from each section, 50 μm apart. One slide was stained for hematoxylin and eosin (H&E) and the other was reserved for immunohistochemistry staining. If no metastatic disease was identified on the first hematoxylin and eosin slide, the reserved slide was stained for pancytokeratin AE1/AE3 (Dako, Glostrup, Denmark). The uterus, ovaries and systematic lymphadenectomies were analyzed according to standard procedures.

### 2.3. Data

Patients’ data were collected prospectively during the study by the investigation team (AT, JA; C.M.S; MR and JR). The statistical analyses were performed with the Statistical Program for the Social Sciences software for Windows (SPSS, version 27.0; SPSS Inc., Chicago, IL, USA).

## 3. Results

From December 2018 to December 2020, 48 consecutive patients with early-stage EC treated in our institution were included in the study. Patient and tumor characteristics are summarized in Table 3.

A minimally invasive approach was undertaken in all patients (52% laparoscopic and 47.9% robotics). In three patients, a minimal transverse laparotomy was performed for uterine extraction due to the large uterine size. One patient required conversion to laparotomy due to hypercapnia and poor tolerance of the Trendelenburg position (78 years old, smoker). No other surgical complications were observed during SLN biopsy or complete lymphadenectomy.

Two different tracers (indocyanine green, ICG and radioactive technetium (Tc) 99 nanocolloid) were injected intracervically in 45 patients. In three patients, only Tc99 was injected due to a laparoscopic near infrared (NIR) camera not being available. In high-risk patients (33), a dual injection (cervically and fundal-myometrium) was carried out. In five of these patients, the fundal injection was performed only with Tc99 due to NIR not being available in two cases, and the other three patients had exclusion criteria from systematic lymphadenectomy.

Almost 1 SLN (pelvic or para-aortic; mean 1.7 (range 1–3)) was detected with Tc99 and/or ICG in all 48 patients. By consequence, the overall detection rate with dual tracers was 100%. Overall pelvic detection rate (ICG or Tc99) was 98%. Bilateral pelvic detection was obtained in 89.5% patients. The overall para-aortic detection rate was 66.7% (22/33). When tracers were analyzed separately, the para-aortic SLN rate was 51.5% (17/33) for the Tc99 tracer and 60.7% for ICG (17/28). This difference was statistically significant (*p* = 0.048) after performing Fisher´s exact test. The incidence of “empty node packets” (absence of nodal tissue in the specimens) was just 2% (1/48) for the ICG tracer and we did not find “empty node packets” in any patient (0%) for the Tc 99 radiotracer (Table 4).

The most frequent anatomical SLN locations in the pelvic area were: external iliac (75%) and obturator (45,8%). Three of 48 patients (6.25%) showed direct atypical drainage outside the standard field of pelvic lymphadenectomy (two presacral SLNs and one with direct drainage to the common iliac artery, all for both tracers). One patient (2%) showed an isolated para-aortic sentinel lymph node with the Tc99 tracer. There were no cases of allergic reactions to any of the tracers.

After the surgery, seven patients (14.6%) were upstaged to stage III because lymph node involvement (Table 5). A total of 6.6% only had pelvic SLN metastasis, 4.2% had pelvic and para-aortic metastasis, and 7.1% (two patients) only had isolated para-aortic nodal metastasis (with negative pelvic lymph nodes). For one of them, final histopathology informed about a fallopian tube serous adenocarcinoma. Then, only one patient (3.5%) had isolated para-aortic metastasis for EC. This patient failed to map any para-aortic SLN, but when we explored the para-aortic area, two enlarged nodes were observed and were removed.

Macroscopic lymph node metastasis was found in six patients (12.5%) and in one case (2%) microdisease in one SLN and isolated cells (ITCs) in contralateral pelvic SLN were found in the final pathology.

Among the four patients in whom pelvic SLNs were positive and a complete para-aortic lymphadenectomy was performed, two of them (50%) also had involvement of para-aortic lymph nodes.

All para-aortic SLNs were negative, and therefore sensitivity of the para-aortic SLN cannot be calculated in our study. The negative predictive value for para-aortic SLN was 100%.

Among the patients in whom no para-aortic SLNs were detected (11), four of them (36.4%) had positive metastases in para-aortic lymphadenectomy.

No metastases were found in non-pelvic SLNs (nodes of complete pelvic lymphadenectomy after SLN was removed). Thus, the sensitivity and negative predictive value (NPV) of SLN pelvic detection was 100%.

In low-risk patients, no lymph nodes metastases were found. Among five intermediate-risk patients, one (20%) presented microscopic disease in pelvic SLNs with negative para-aortic lymphadenectomy. Among 33 high-risk patients, six had lymph nodes metastasis (18.2%), four of them were pelvic-SLNs, and the other two had macroscopic nodes removed from the para-aortic lymphadenectomy as para-aortic SLN was not detected for them.

## 4. Discussion

Currently, some practice guidelines consider SLN biopsy as an acceptable alternative to systematic lymphadenectomy for surgical staging of patients with pre-operative stage I/II EC [4,5,14]. Several prospective cohort trials have shown high sensitivity to detect pelvic nodes metastases and a high negative predictive value by applying a sentinel lymph node algorithm on lower intermediate-risk and high-risk endometrial carcinomas [6,15]. ICG is the most recommended tracer in research and guidelines [16], but most of them insist on the importance of experienced surgeon teams.

It has been shown that the learning curve for successful SLN mapping stabilizes after 30–40 cases [17,18]. When starting this trial, our group had no experience with the use of ICG. Instead, we had worked extensively with the Tc99 tracer in cervical cancer. According to the importance of having a correct learning curve, injecting both tracers could help us reach security in detecting ICG SLN. The discussion with nuclear medicine specialists of SPECT-CT images gave information to the surgeon before the surgery. Moreover, during surgery when ICG SLN was detected, the gamma probe allowed confirmation with Tc99 radioactivity. With this double approach, we obtained a high overall pelvic DR (98%) and even bilateral DR (89.5%) also in the first 20 cases. These DR were higher than those achieved in a FIRES trial [6] (overall DR of 86%, bilateral only 52%), in which criticism was directed at the low experience in the use of ICG of some surgeons. In our series, when pelvic SLNs were detected, the remaining non-SLNs were negative in all patients. From our point of view, this could confirm the accuracy of the multidisciplinary approach during learning curve. Dual mapping approach has been described as a better approach in situations of high complexity such as in SLN after neoadjuvant treatment in patients with breast cancer [19,20].

We agree with other authors [18] that to reach a successful SLN mapping it would be enough to confirm the presence of lymph nodes in the biopsied tissue and would not be necessary to complete the lymphadenectomy to validate the SLN technique. We could have saved 79.1% (38/48) of systematic pelvic lymphadenectomy and 60.7% (17/28) of para-aortic lymphadenectomy.

Just one surgical specimen (2%) believed to be SLN was found not to be any lymphatic tissue on pathological analysis (“empty node packet”). This tissue was green (ICG) but intra-operatively “cold” for Tc99. This rate is lower than that found by other authors. The FILM trial reported an “empty node packet” rate of 5% using only ICG [21], and Cabrera et al. reported the rate of the absence of nodal tissue to be 4% using the combination Tc99 with ICG [22] and 0% using Tc99-methylene blue. In our study, injecting Tc99 and ICG in the same patient could help us better detect “real” lymph nodes in “green” tissue.

An important highlight in our study is the high SLN pelvic rate, as well as a high bilateral rate without the need, in this series, for cervical reinjection as described by other authors [15,23].

In our study, seven patients (14.6%) were upstaged to stage III for lymph node involvement. This has clinical implications, and these results are similar to the literature [7]. The patients who most benefited from the SLN technique (with systematic para-aortic lymphadenectomy when SLN mapping failed) were intermediate–high-risk. In these patients the upstaged rate was 18.2% (6/33).

Superficial and deep cervical injection is the most common and simplest injection technique used in multiple trials [6,23] and has been settled as a standardized technique for pelvic staging [4]. Exclusively cervical injection is associated with low aortic SLN detection rates, between 11% [24] and 23% [6]. This rate increases up to 59.4% when an intra-operative ICG fundal injection is added [13]. Torné et al. [12] described the TUMIR method using transvaginal ultrasound-guided myometrial injection of radiotracer, obtaining a para-aortic DR of 45.4% using only Tc99 and 34% using a hybrid tracer (ICG–Tc99) [25]. In our series, in high-risk patients, a double injection (cervical and fundal) was performed with both tracers separately and the overall para-aortic DR was 66,7%. The para-aortic SLN rate was 51.5% (17/33) for the Tc99 tracer and 60.7% for ICG (17/28). This difference could be explained due to the injection technique being different (intra-operative fundal through the endocervical canal for ICG versus the pre-operative TUMIR technique for Tc99). The advantage of the TUMIR technique is that we can know if SLN is mapped pre-operative (with SPECT-CT), but it is more unpleasant for the patient. Intra-operative fundal ICG injection seems to have higher DR in experimented teams and has the advantage that it is simpler and more convenient for the patient. When dual tracer (ICG and Tc99 in separate injections) is used, DR seems to be higher. Hybrid tracer (ICG and Tc99 in the same injection) would possibly summarize the advantages of both tracers: Tc99 would fix ICG avoiding the staining of the rest of the lymph nodes if, for any reason, the surgeon takes longer to dissect the lymph node area, and it could be injected intra-operatively. However, a recently Spanish retrospective study concluded that combined use of ICG and Tc99 did not improve SLN detection rates in endometrial cancer [22] although randomized prospective studies would be necessary. Sanchez-Izquierdo et al. reported a low overall detection rate (69%) and a bilateral pelvic detection of 56% using a hybrid tracer (ICG–Tc99) injected using the TUMIR approach [25].

The effort to improve para-aortic DR has been in order to minimize the possibility of missing occult isolated para-aortic lymph nodes (with negative pelvic SLNs). However, the incidence of isolated para-aortic lymph node (IPL) metastasis ranges from 1% to 3% [13]. Anatomical study has proven that EC can directly metastasize into the para-aortic area through the pelvic–infundibular ligament pathway. The dissection of the para-aortic area is left to the surgeon’s decision in the NCCN SLN algorithm [5] and in the ESGO/ESTRO/ESP guidelines [4]. Kim et al. [26] showed that a sequential injection of ICG in the bilateral uterine corpus followed by cervical injection (two-step group) improved the para-aortic SLN detection rate from 38.2% to 57% in the upper para-aortic area (*p* < 0.001), and from 18.7% to 67.1% in the lower para-aortic area (*p* < 0.001). The one-step group (only cervical injection) had 2.5% of IPL metastasis while there were no missed para-aortic lymph node metastases in the two-step group.

In our study, two patients (7.1%) had IPL metastasis, though one of them had synchronic endometrial and ovarian cancer, and the para-aortic metastasis could be due to ovarian cancer, so the DR for IPL would be 3.5% (1/28). However, in these two cases, para-aortic SLNs were not detected (mapping failed). Para-aortic lymphadenectomy was performed because some macroscopic suspicious lymph nodes were observed when the para-aortic area was explored. Thus, in high-risk patients in whom para-aortic SLN failed, proper evaluation of aortic nodal status would assist in the tailoring of adjuvant and prevent undertreatment of patients with isolated para-aortic metastasis.

On the other hand, when pelvic lymph nodes were positive, 51% had para-aortic metastases [7]. We found very similar results (50%). In our series, for all patients with pelvic SLN metastases and para-aortic metastases, the para-aortic SLN was not identified, neither by ICG nor by Tc-99m. We can assume that it was possibly caused by inhibited tracer flow due to lymphatic obstruction by the tumor (tumor lymphatic blockage). All these patients were in the intermediate–high-risk EC group.

Due to the small sample size of our study, no strong conclusions can be drawn, but it seems that although we have a higher rate of para-aortic detection, this would not substantially change clinical management. This issue should be further investigated. The detection of the para-aortic SLNs would be useful in those cases in which the pelvic nodes are positive, and we would make sure to extract the para-aortic node with a greater risk of metastasis. In cases where there was no migration, we would have to complete the lymphadenectomy.

To our knowledge, there is currently no prospective study in the literature that injects a double tracer (ICG and Tc99) and double site injection (cervical and fundal) in the same group of patients. This is one of the strengths of our study

## 5. Conclusions

Dual tracers (ICG and Tc99) could help increase the accuracy of the SLN biopsy technique during the learning curve. Double injection (cervical and fundal) with double tracers offers good overall detection rates and increases para-aortic SLN detection, but in our series, does not allow us to find more metastatic para-aortic nodes. When pelvic SLNs are affected, the para-aortic area should be carefully assessed.

## Figures and Tables

**Figure 1 cancers-14-00929-f001:**
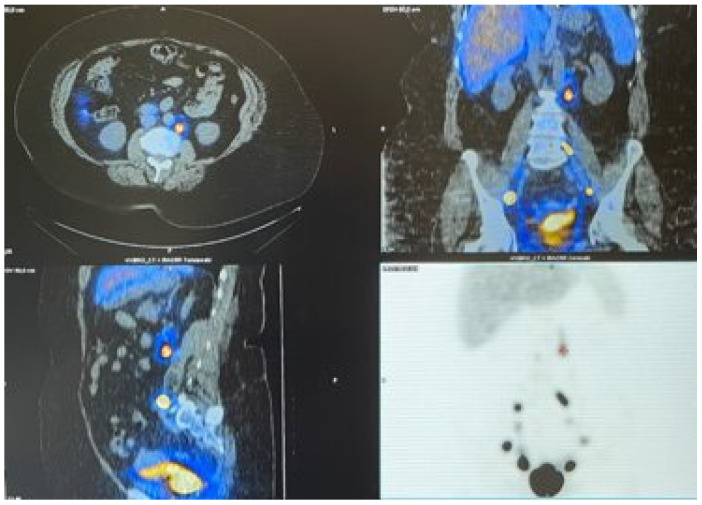
SPECT/CT with pelvic and para-aortic mapping.

**Figure 2 cancers-14-00929-f002:**
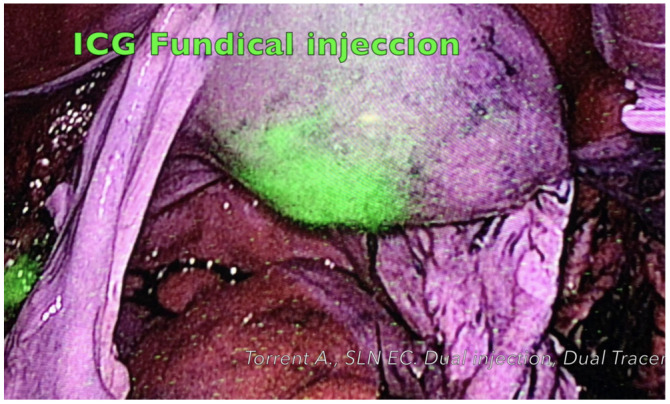
ICG intra-operative fundal injection.

**Table 1 cancers-14-00929-t001:** Exclusion criteria.

- Contraindication for surgical treatment
- Suspicion of metastasic disease
- Pathological pelvic or para-aortic lymph nodes in CTscan or MRI (no FNA needed)
- BMI > 45 (for para-aortic SLN detection)
- Story of radiotherapy in the pelvic or para-aortic regions
- Age > 80 years old (for para-aortic lymphadenectomy) or < 18 years old
- Thyroidal hyperfunction
- Clinical Frailty Scale > 5 (for para-aortic SLN detection)

BMI: Body mass index; CTscan: computed tomography scan; MRI: magnetic resonance imaging.

**Table 2 cancers-14-00929-t002:** Planned surgery according to ESGO/ESTRO/ESP recommendations: LVSI, lymphovascular space invasion; BSO, bilateral salpingo-ooforectomy; SLN, sentinel lymph node dissection; PLA, pelvic lymphadenectomy; PALA, para-aortic lymphadenectomy.

Risk Group	Molecular Classification Unknown	Injection (Tc99 + ICG)	Planned Surgery
Low	Stage IA endometrioid + low grade + LVSI negative o focal	Cervical	Hysterectomy BSO Pelvic SLN
Intermediate	Stage IB endometrioid + low grade + LVI negative o focal Stage IA endometrioid+ high-grade + LVSI negative or focal Stage IA non-endometrioid (serous, clear cell, undifferentiated carcinoma, carcinosarcoma, mixed) without myometrial invasion	Cervical	Hysterectomy BSO Pelvic SLN PLA
High-intermediate	Stage I endometrioid + substancial LVSI regardless of grade and depth of invasion Stage IB endometrioid high-grade + regardless of LVSI status Stage II	Cervical Fundal	Hysterectomy BSO Pelvic SLN PLA Para-aortic SLN PALA

**Table 3 cancers-14-00929-t003:** Patients and tumor characteristics.

Characteristics	Patients (*n* = 48)
Age (years)	63.5 (47–78)
BMI (kg/m^2^)	33.2 (20–49)
Histologic type (*n* (%))	
Endometrioid	33 (68.7%)
Serous	10 (20.8%)
Mixed	1 (2%)
Carcinosarcoma	2 (4.1%)
Clear cell	2 (4.1%)
Tumor Grade (*n* (%))	
G1	10 (20.8%)
G2	16 (33.3%)
G3	22 (45.8%)
Miometrial invasion	
No	1 (2%)
<50%	26 (54.2%)
>50%	21 (43.7%)
Lymphovascular invasion (*n* (%))	13 (27%)
Surgical approach	
Laparoscopy	25 (52%)
Robotics	23 (47.9%)

**Table 4 cancers-14-00929-t004:** Sentinel lymph nodes’ detection rates.

SLN Detection Rate	Number of Patients (%)		
	ICG	Tc 99	ICG or Tc99
Overall pelvic detection	45/45 (100)	45/48 (94)	47/48 (98)
Unilateral pelvic detection	4/45 (8.8)	5/48 (10)	4/48 (8.3)
Bilateral pelvic detection	41/45 (91)	40/48 (83)	43/48 (89.5)
Para-aortic detection	17/28 (61)	17/33 (52)	22/33 (66.7)
SLN atypical drainage	3/45 (6.6)	3/48 (6.2)	3/48 (6.2)
Not detected (pelvic)	0/45 (0)	3/48 (6.2)	1/48 (2)
Not detected (para-aortic)	11/28 (39.3)	16/33 (48.5)	11/33 (33.3)
“empty node packet”	1/48 (2)	0/48 (0)	1/48 (2)

**Table 5 cancers-14-00929-t005:** Lymph node metastases (7/48 patients): PLA, pelvic lymphadenectomy; PALA, para-aortic lymphadenectomy; +, positive histopathology; -, negative histopathology.

Case	Age	Initial Stage	Pelvic SLN		PLA	Para-Aortic SLN	PALA	ILV	Final FIGO Stage
			Tc99	ICG					
1	75	IaG3 (Serous)	+ Unilat (1/2)	+ Unilat (1/2)	-	No injection (IMC 47)	Not done	+	IIIC1
2	67	Ia G3 (Serous)	+ Unilat (1/2)	+ Unilat (1/2)	-	No detected	-	+	IIIC1
3	56	Ia/b G1 (Endometrioid)	+ R (itc) + L (micro)	+ R (itc) + L (micro)	-	No injection	-	+	IIIC1 G2
4	69	IbG3 (Endometrioid)	+ R (1/3) + L (2/2)	− R (0/2) + L (2/2)	-	No detected	+ (2/8)	+	IIIC2
5	52	IbG3 (Endometrioid)	+ Unilat (1/2)	+ Unilat (1/2)	-	No detected	+ (2/9)	+	IIIC2
6	78	IaG3 (Serous)	-	-	-	No detected	+ (2/6)	+	IIIC (Fallopian Tube)
7	75	IbG3 (Serous)	-	-	-	No detected	+ 2 bulky	+	IIIC2

## Data Availability

The data presented in this study are not openly available due to confidentiality reasons but are available upon reasonable request from the corresponding authors.

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
