# Peer review of "Sentinel Lymph Node Biopsy in Endometrial Cancer: Dual Injection, Dual Tracer—A Multidisciplinary Exhaustive Approach to Nodal Staging"

_cancers, 2022, doi:10.3390/cancers14040929_

Round 1

Reviewer 1 Report

Dear Authors, I am pleased to read your prospective observational single-center study that gives a slightly different way to detect the pelvic and para aortic SLN. 

In high risk cases you did fundal ICG injection first, para-aortic SLN retrieval(ICG plus by gamma probe), then systematic para-aortic lymphadenectomy. This was followed by intracervical ICG injection and then retrieval of pelvic SLN. 

From the methods, it appears that the surgeon has retrieved the para-aortic sentinel before the cervical injection. So the second additional injection would not have added anything substantial to the aortic sentinel detection. (conclusion mentions that dual injection increases para-aortic SLN detection , please clarify)

There is no mention of the time it takes to do this injection procedure, para-aortic lymphadenectomy,  and can this extra time affect the mapping. . And because of this extra time, T99 hot nodes help in identifying the real SLN.

As per the guidelines, a successful mapping after cervix injection is defined by observing a channel leading from the cervix directly to at least one candidate lymph node in at least one hemipelvis. Your study uses an additional fundal injection. And in order to know the contribution of fundal injection to pelvic SLN detection, it would have been interesting to know the number of times the lymphatic channel from the fundus to SLN node was identified. 

I agree that a dual tracer would help in detecting the sentinel node in the learning curve, as SPECT/CT would provide the region of the sentinel node, especially if the lymphatic channel is not mapped. I, however, could not understand what is the contribution of cervical injection for the detection of para-aortic sentinel nodes and the contribution of fundal injection for pelvic LN detection.

Table 3, the SLN detection rate by ICG is better than Tc 99 in all scenarios. It would be better to discuss your suggestion, on who should use dual tracer and why.

All 4 para-aortic positive had failed mapping of para-aortic region and all negative para-aortic were mapped. It would be interesting to discuss the reason, for failed mapping by the fundal injection and dual tracer. 

The discussion can be shorted and made crisper.

In the end, I would say that it is a good study and will help to pave the way for mapping the aortic nodes in EC in the future. 

Best Wishes,

Reviewer 2 Report

This work is well written.

Nevertheless, the objectives, which are too numerous and not very precise, should be better clarified: what the authors are finally trying to evaluate is the performance of the ICG/Tc99m couple and not individually, which has already been widely published in the literature.

Therefore, the results should be written in this sense with more clarity to facilitate the reading and the discussion refocused on this topic.

For the benefit of readability, I will concordance the new recommendations of Concin et al. of 2021 in the tables (endometrioid/non-endometrioid, low grade/high grade).

In the discussion, I will further qualify the interest of the ICG-Tc99m association, the authors can also quote PMID: 31992599

The discussion is very long: I don't see the interest of the histological analysis in this study.

The conclusion should be reworded by the authors' objectives (Tc 99 vs ICG already demonstrated) but the interest of this association for bilateral pelvic and para-aortic detection

Round 2

Reviewer 2 Report

The authors have modified their article according to the comments previously sent. This clearly clarified their study and met their objectives.